# Close-packed polymer crystals from two-monomer-connected precursors

Hong-Joon Lee[1], Yong-Ryun Jo[1], Santosh Kumar[1], Seung Jo Yoo[2], Jin-Gyu Kim[2], Youn-Joong Kim[2], Bong-Joong Kim[1] & Jae-Suk Lee[1]

The design of crystalline polymers is intellectually stimulating and synthetically challenging, especially when the polymerization of any monomer occurs in a linear dimension. Such linear growth often leads to entropically driven chain entanglements and thus is detrimental to attempts to realize the full potential of conjugated molecular structures. Here we report the polymerization of two-monomer-connected precursors (TMCPs) in which two pyrrole units are linked through a connector, yielding highly crystalline polymers. The simultaneous growth of the TMCP results in a close-packed crystal in polypyrrole (PPy) at the molecular scale with either a hexagonal close-packed or face-centred cubic structure, as confirmed by high-voltage electron microscopy, and the structure that formed could be controlled by simply changing the connector. The electrical conductivity of the TMCP-based PPy is almost 35 times that of single-monomer-based PPy, demonstrating its promise for application in diverse fields.

[1] School of Materials Science and Engineering, Gwangju Institute of Science and Technology (GIST), 123 Cheomdangwagi-ro, Buk-gu, Gwangju 61005, Korea. [2] Center for Electron Microscopy Research, Korea Basic Science Institute (KBSI), 169-148 Gwahak-ro, Yuseung-gu, Daejeon 34133, Korea. Correspondence and requests for materials should be addressed to J.-S.L. (email: jslee@gist.ac.kr).

The polymers of today are required to be more than simply flexible. They must also be able to transfer and store particles such as electrons[1], ions[2] and gases[3]. Possession of these qualities allows polymers to be used in state-of-the-art devices as (semi)conductors[4,5], electrode-active materials[6] and photo-active materials[7], as well as in many more capacities. Numerous attempts have been made to satisfy these requirements by introducing functionalities on the backbone or side chains of a polymer. However, this approach has yielded unsatisfactory results because of the inherent morphological limitations of polymers, which result in poor transferability[8]. Many attempts have been made to enhance the crystallinity of polymers using various synthetic and processing methods, but the formation of one-dimensional chains induces random entanglements, which result in large amorphous regions[9].

Recently, not only single-crystal polymers[10,11] at the scale of hundreds of nanometres but also covalent organic frameworks[12,13] and two-dimensional polymers[14,15] have been developed as new concepts for polymers with well-ordered structures, and these developments have encouraged research on crystalline polymers. The primary difficulty encountered in the study of crystalline polymers is the control of the precise arrangement of the molecules that contain functionalities, which requires the use of expensive and complex synthetic methods[8,16], resulting in the loss of one of the advantages and core principles of polymer engineering, namely, the development of materials at a reasonable cost. To address this challenge, several efforts have been made[17–19] to synthesize crystalline polymers via simple processes and to study their crystal structures using high-resolution transmission electron microscopy (HRTEM) and X-ray diffraction. Nevertheless, beyond the mere existence of the polymer lattice, polymer science has yet to definitively characterize the three-dimensional crystal structures of polymers because of the difficulties encountered in the formation and characterization of polymer crystals. To the best of our knowledge, unlike in metals and ceramics, which commonly exhibit a primary metallic crystalline structure[20] consisting of hexagonal close-packed (HCP) and face-centred cubic (FCC) structures, there have been no reports of such structures in polymers, either natural or synthetic.

To maximize the efficiency of an existing polymer, extensive research is needed to synthesize and evaluate its three-dimensional structure at the molecular scale, just as in the case of metals and ceramics. Therefore, we produced highly crystalline conjugated polymers using two-monomer-connected precursors (TMCPs), in which two monomer units are linked through a connector, yielding crystalline structures at the molecular scale while inhibiting the entropically favourable chain entanglements. For the design of the TMCPs, we selected conventional pyrrole (Py) as the monomer and various disulfonic acids (DSAs) to serve as both a connector and a dopant. Here the role of the dopant is to endow the polymer crystal structure with conductivity. We used high-voltage electron microscopy (HVEM), which is capable of measuring quantitative structural data along different zone axes, to explore the crystal structures of the synthesized polymers. We identified a primary metallic crystalline structure at the molecular scale in the P(Py:DSA:Py)s (where a P(Py:DSA:Py) is a polymer synthesized from a TMCP composed of Py and a DSA) that could manifest as either HCP or FCC; the structure that formed could be controlled by simply changing the connector. The electrical conductivities of the P(Py:DSA:Py)s with HCP and FCC structures were greatly increased compared with that of single-monomer-based polypyrrole (PPy), demonstrating the promise of these materials for application in diverse fields such as organic electronics, sensors and energy storage technology. Our results regarding our polymerization strategy and morphological analysis in three dimensions can serve as the basis for new approaches to polymer crystallography.

## Results

**Polymerization of the TMCPs.** When polymerization proceeds with a TMCP, unlike for a single monomer, the four reactive sites of the TMCP impose a constraint on the direction of propagation, thereby preventing randomly oriented chain growth (Fig. 1a). The polymerization process of a TMCP, as described here, is based entirely on a chemical oxidation reaction on the substrate, enabling facile one-step synthesis (Supplementary Fig. 1). This type of in situ polymerization on various substrates has been previously reported by others[21–23]. Before polymerization, the Py monomer was reacted with 1,2-ethanedisulfonic acid (EDSA), 1,4-butanedisulfonic acid (BDSA) or 4,4'-biphenyldisulfonic acid (BPDSA) to form a TMCP (Py:EDSA:Py, Py:BDSA:Py or Py:BPDSA:Py) (Fig. 1b,c). Methanesulfonic acid (MSA) was used to prepare a single-monomer-based precursor (Py:MSA) as a reference. The ionic bonding that formed as a result of the acid–base interaction in all precursors was easily confirmed via Fourier transform infrared (FT-IR) spectroscopy by observing the absence of N–H and C=C stretching in the neutral Py ring[24] (Supplementary Fig. 2). All TMCPs (Py:DSA:Py) were polymerized at ~0 °C for 12 h in the presence of an oxidant, leading to the formation of uniform films on various substrates (Supplementary Fig. 3).

Ultraviolet–visible absorption spectra clearly indicated that the P(Py:DSA:Py) films had been successfully synthesized, exhibiting an absorption peak at ~430 nm due to a polaron band of p-doped PPy (ref. 25; Supplementary Fig. 4). Interestingly, P(Py:EDSA:Py), P(Py:BDSA:Py) and P(Py:BPDSA:Py) exhibited redshifted absorption spectra compared with that of P(Py:MSA). Such redshifted absorption indicates the existence of an electronic interaction between the stacked polymer chains, which causes the energy bandgap to be smaller than that of an amorphous polymer with the same polymer backbone structure[26–28]. Furthermore, the degree of connection established via ionic bonding was confirmed through X-ray photoelectron spectroscopy (Supplementary Fig. 5). In the nitrogen ($N_{1s}$) signals, two components were observed that are indicative of $–NH^{\bullet\,+}–$ (~400.2 eV) and $=NH^{+}–$ (~401.6 eV) in the polaron and bipolaron states, respectively[29]. This finding confirmed that the Py units were linked by DSA connectors.

**Molecular crystal structures in the P(Py:DSA:Py)s.** We analysed the crystalline structure of the synthesized P(Py:DSA:Py)s using high-voltage (1,250 kV) electron beams. Such electron beams have the ability to acquire quantitative structural data at the sub-nanometre scale at high voltage. The unconnected P(Py:MSA) demonstrated an amorphous nature in HRTEM and X-ray diffraction analyses (Supplementary Fig. 6), whereas for P(Py:BDSA:Py), although the X-ray diffraction pattern exhibited two characteristic peaks, low crystallinity and small crystal domains were observed, making analysis of the crystal structure difficult (Supplementary Fig. 7). This can be reasonably interpreted to indicate that the rotational freedom of the butyl group of BDSA hindered the crystal packing. By contrast, the P(Py:EDSA:Py), produced from a rigid TMCP, exhibited a highly crystalline structure (Fig. 2a). From a detailed analysis of the HRTEM images and their fast Fourier transforms (FFTs) observed along different zone axes (z), we demonstrated that the P(Py:EDSA:Py) possessed an HCP crystal structure[30] with a space group of $P6_3/mmc$ (ref. 31; Supplementary Fig. 8).

Figure 2a shows the HRTEM images and the corresponding Fourier-mask-filtered FFT images of P(Py:EDSA:Py) observed

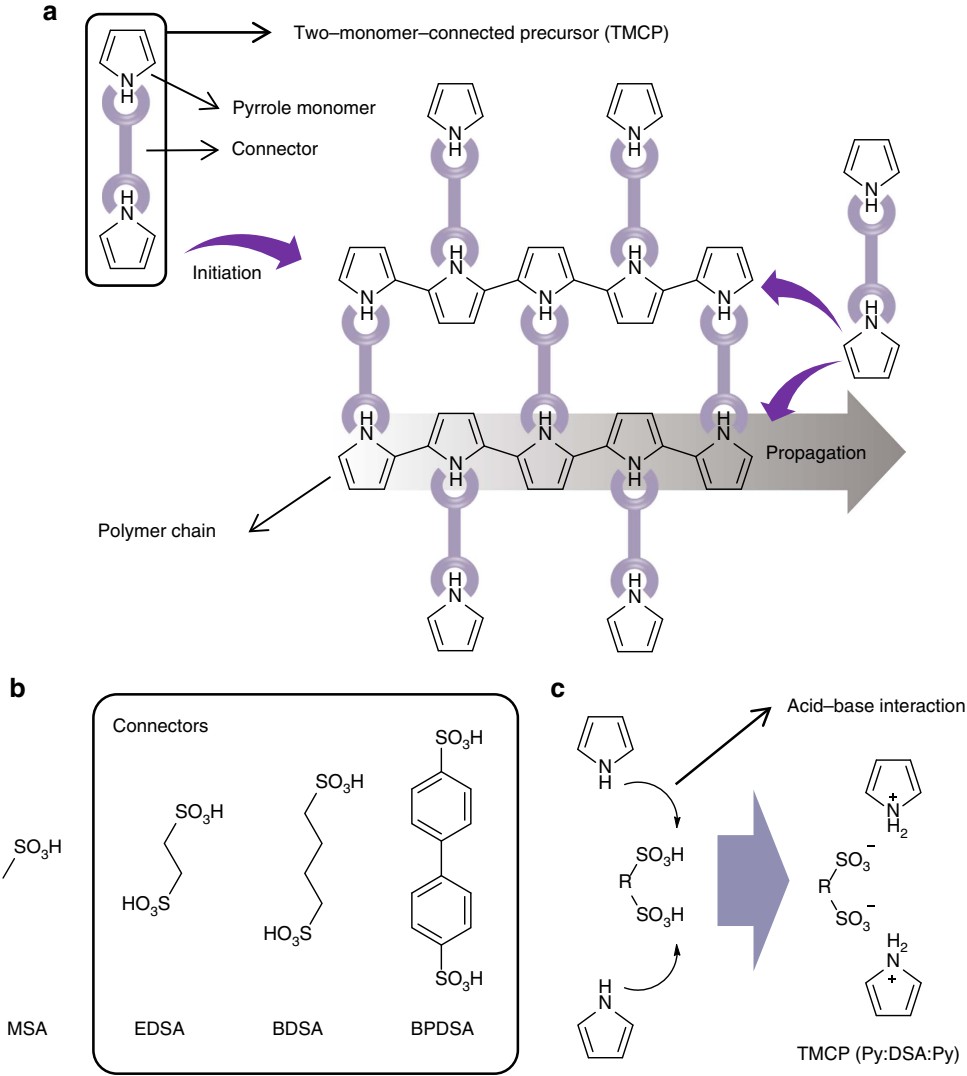

**Figure 1 | Schematic illustration of the polymerization of a TMCP.** (**a**) Chemical oxidative polymerization of a TMCP, avoiding chain entanglement. (**b**) Molecular structures of the connectors. (**c**) Acid–base interaction forming ionic bonds between the Py and connectors.

along different beam directions with respect to the HCP structure ($z = [\bar{2}4\bar{2}3]$, $[\bar{1}2\bar{1}6]$ and $[1\bar{2}13]$). All HRTEM images have $\{10\bar{1}0\}$ in common, and their inter-planar spacing ($d$ spacing) is $\sim 0.41$ nm, from which a lattice parameter $a$ of 0.47 nm for P(Py:EDSA:Py) can be easily calculated[32] (Fig. 2b). From the lattice parameter $a$, we also calculated the (0002) $d$ spacing, which is the second-order reflection of the lattice parameter $c$ and corresponds to the X-ray diffraction peak at $2\theta = 23.8°$, in accordance with the calculated length (0.37 nm) between repeating units[33] (Figs 2b and 3a). Considering the packing sequence of the HCP crystals[32], P(Py:EDSA:Py) is assumed to have grown along the [0001] direction, as the lattice parameter $c$ is twice the distance between two repeating units. The regions of dark contrast in the HRTEM images represent clusters of ionic bonding, with a higher electron density between the ammonium and sulfonate than that in carbon[34]. Therefore, we can reasonably conclude that the ammonium–sulfonate clusters were hexagonally packed in the (0001) plane, with a unit cell with lattice parameters of $a = 0.47$ nm and $c = 0.74$ nm, corresponding to a $c/a$ ratio that is consistent with the primitive unit cell of a typical HCP crystal[32] (Supplementary Fig. 10a).

Interestingly, when the BPDSA connector was used, the crystal structure changed from HCP to FCC. An FFT analysis of the

P(Py:BPDSA:Py) revealed that the crystal structure corresponded to an FCC crystal[30] (Supplementary Fig. 9). Figure 2c shows HRTEM images and the corresponding filtered FFT images of P(Py:BPDSA:Py) observed along different zone axes ($z = [001]$, $[011]$ and $[\bar{1}11]$). Notably, the (400) $d$ spacing, which is the fourth-order reflection of the $d$ spacing (1.48 nm), is consistent with the calculated length (0.37 nm) between repeating units[33]. The X-ray diffraction spectrum was also used to investigate the $d$ spacings in the [100] direction, and peaks were observed at $2\theta = 6.1°$ (1.48 nm), 12.2° (0.74 nm), 18.2° (0.49 nm), 24.4° (0.37 nm) and 30.4° (0.29 nm) (Fig. 3b). Considering the crystal stacking sequence of the FCC structure along the [111] direction[32], the BPDSA connectors, in which the length between sulfonate groups is 1.05 nm, were concluded to be diagonally packed in between PPy chains, as shown in Supplementary Fig. 10b. The $d$ spacings of P(Py:BPDSA:Py) are relatively long because two different types of ammonium–sulfonate ionic clusters exist (grey and blue circles in Supplementary Fig. 10b), which have different ionic bond angles and lengths (and consequently slightly different electron densities) because of the diagonally packed BPDSA connectors between the PPy chains. Therefore, the molecular crystal structure in P(Py:BPDSA:Py) is similar to the layered rock-salt

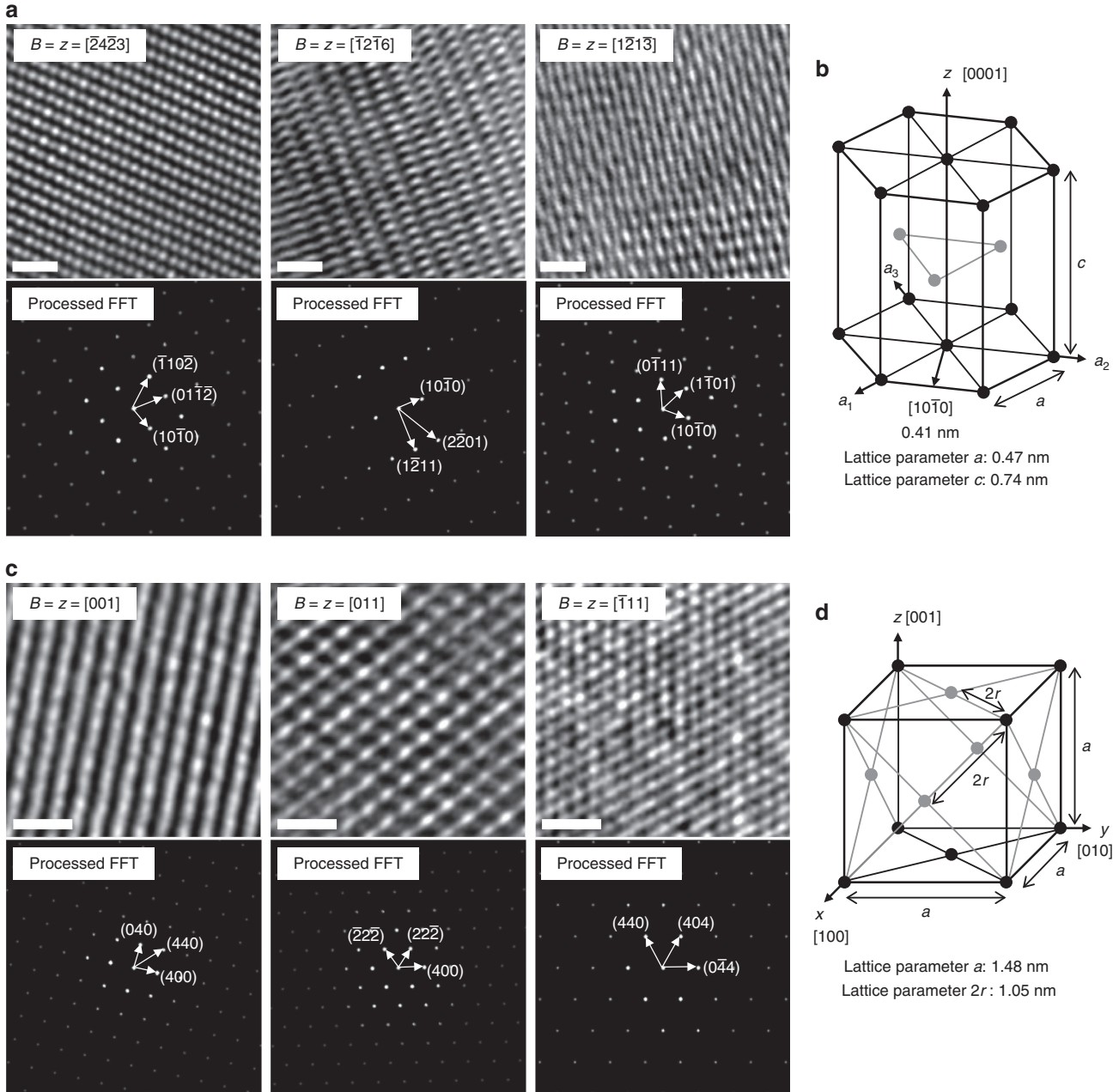

**Figure 2 | Characterization of the molecular crystal structures of P(Py:EDSA:Py) and P(Py:BPDSA:Py). (a)** HRTEM and processed FFT images of P(Py:EDSA:Py) measured along different zone axes ($z = [\bar{2}4\bar{2}3]$, $[\bar{1}2\bar{1}6]$ and $[1\bar{2}1\bar{3}]$). Scale bar, 1 nm. **(b)** A reduced-sphere unit cell of P(Py:EDSA:Py), illustrating the HCP crystal structure. **(c)** HRTEM and processed FFT images of P(Py:BPDSA:Py) measured along different zone axes ($z = [001]$, $[011]$ and $[\bar{1}11]$). Scale bar, 1 nm. **(d)** A reduced-sphere unit cell of P(Py:BPDSA:Py), illustrating the FCC crystal structure.

structure[35] (space group $Fm\bar{3}m$) in terms of the ordering of the different ammonium–sulfonate clusters. Furthermore, we determined the unit cell of P(Py:BPDSA:Py), with lattice parameters of $a = 1.48$ nm and $2r = 1.05$ nm, corresponding to an $a/2r$ ratio that is consistent with a typical FCC crystal[32] (Fig. 2d).

Unlike in covalent bonding, in the case of ionic bonding in solution, the bond length and angle between the polymer chain and the connector can be influenced during self-assembly by factors in the surrounding environment, such as counter ions and steric hindrance. We obtained molecular-level-ordered polymers with different thermodynamically stable structures depending on the connector used. These polymers formed the most stable crystal structures (HCP and FCC) possible in each case, as both are close-packed structures.

**Electrical conductivity of the P(Py:DSA:Py) films.** To understand the effect of the molecular crystals formed in the synthesized polymers on the electron transferability, we measured the electrical conductivity of P(Py:DSA:Py) films with thicknesses of ∼150 nm using the van der Pauw method (Fig. 4). The electrical conductivity of P(Py:EDSA:Py) (8.94 S cm$^{-1}$) was almost three times higher than that of P(Py:MSA) (3.04 S cm$^{-1}$). In the case of P(Py:BDSA:Py), the conductivity was higher than that of P(Py:MSA) but lower than that of P(Py:EDSA:Py); this finding can be attributed to their relative crystallinities. In particular, the conductivity of P(Py:BPDSA:Py) (103.7 S cm$^{-1}$), which was formed using an aromatic connector, showed a marked increase compared with that of P(Py:MSA). These results suggest not only that the crystal structure strongly affects the

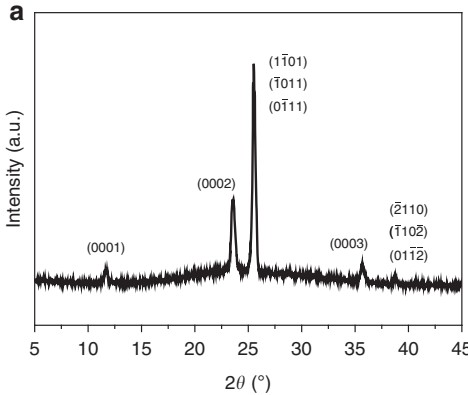

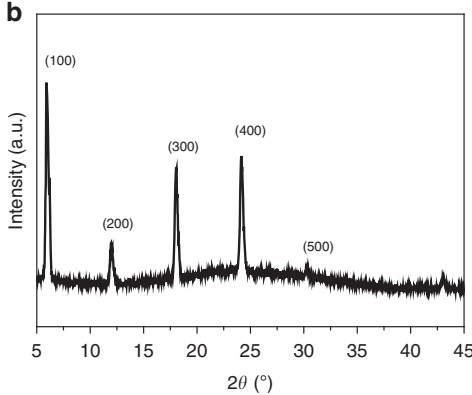

**Figure 3 | X-ray diffraction spectra of P(Py:EDSA:Py) and P(Py:BPDSA:Py) films.** The measured lattices identified via HRTEM were also confirmed through X-ray diffraction analysis. (**a**) X-ray diffraction spectrum of the P(Py:EDSA:Py) film. The *d* spacings in the [0001] direction observed in the X-ray diffraction spectrum are consistent with the lattice parameter *c* and the theoretical length between repeating units. (**b**) X-ray diffraction spectrum of the P(Py:BPDSA:Py) film. The *d* spacings in the [100] direction observed in the X-ray diffraction spectrum are consistent with the lattice parameter *a* and the theoretical length between repeating units.

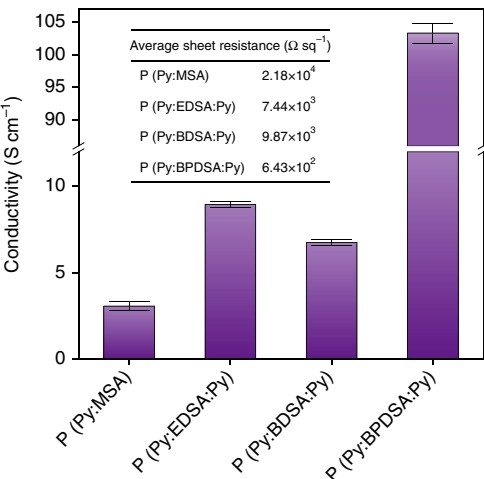

**Figure 4 | Electrical conductivities of P(Py:MSA) and P(Py:DSA:Py) films.** The electrical conductivities of the synthesized polymer films measured using the van der Pauw method at room temperature. The bar graphs show the mean conductivity measured for three samples of each polymer, with error bars representing the s.d. (inset: the average sheet resistance values of each film).

electrical conduction pathways but also that the aromatic structure of the biphenyl group in the connector enhances the charge transfer along the connector by virtue of its own $\pi$-conjugation system.

## Discussion

In this work, we synthesized P(Py:DSA:Py)s via the polymerization of TMCPs to achieve molecular-level crystallinity. HVEM analysis was performed to obtain quantitative structural data at the sub-nanometre scale using high-voltage electron beams. The polymers were found to possess long-range-ordered structures consisting of two different close-packed crystal structures (HCP and FCC) at the molecular scale. Polymerization from a TMCP results in a considerable increase in electrical conductivity because of the close-packed crystal structures of P(Py:DSA:Py)s. Notably, the conductivity of P(Py:BPDSA:Py), which is formed using an aromatic connector, is markedly increased by a factor of more than 35 compared with that of unconnected PPy. The polymerization method and analysis strategy applied in this study can be extended to other functional polymers using various TMCPs for applications such as organic electronics and electrochemical devices, and this work can also serve as the basis for new approaches to polymer crystallography.

## Methods

**Materials.** Unless otherwise stated, all materials were purchased from commercial suppliers and were used as received. Glass substrates (glass slides, 76 × 26 mm) were purchased from Marienfeld-Superior. Indium tin oxide-coated glass slides (15–25 $\Omega$ sq$^{-1}$, 75 × 25 mm) were purchased from Aldrich. Poly(ethylene terephthalate) films (thickness: 100 $\mu$m) for use as flexible and transparent substrates were purchased from Film Bank. Flexible graphite foils (thickness: 130 $\mu$m) were purchased from Dongbang Carbon.

**Preparation of P(Py:MSA) and P(Py:DSA:Py) films.** First, 9.80 mg of MSA (Sigma-Aldrich), 9.50 mg of EDSA (Tokyo Chemical Industry), 11.8 mg of BDSA (Ark Pharm) and 15.7 mg of BPDSA (Tokyo Chemical Industry) were each individually dissolved in 5 ml of deionized water in 20 ml capped vials. Then, 7.00 $\mu$l of Py (Tokyo Chemical Industry) was added to each vial. The mixtures of Py and acid were stirred for 2 h at $\sim$0 °C to prepare the TMCPs. The substrates (glass slides, indium tin oxide-coated glass slides, poly(ethylene terephthalate) films or graphite foils) were successively ultrasonicated with deionized water, acetone (Sigma-Aldrich) and isopropyl alcohol (Aldrich) for 15 min each. Clear substrates were vertically immersed in each vial. The washed substrates were then dried in an oven at 60 °C, and their surfaces were treated by means of combined exposure to ultraviolet and ozone. The *in situ* chemical oxidative polymerization of P(Py:MSA) and P(Py:DSA:Py) was induced on the substrates at $\sim$0 °C by adding 5 ml of a 0.10 mmol ammonium persulfate (Aldrich) aqueous solution. After 12 h, uniform P(Py:MSA) and P(Py:DSA:Py) films were obtained on the substrates. The films were rinsed with deionized water and ethanol (Aldrich) to remove any remaining salts and unreacted monomers. The washed films were then dried in a vacuum oven at 80 °C overnight.

**Characterizations of TMCPs and P(Py:DSA:Py)s.** The FT-IR spectra of the TMCPs were measured using a Perkin-Elmer FT-IR spectrometer (Spectrum System 2000) with potassium bromide pellets (Fluka). Liquid Py was prepared by forming a liquid capillary film between two potassium bromide pellets. The ultraviolet–visible spectra of P(Py:MSA) and P(Py:DSA:Py) films that were *in situ* polymerized on glass were obtained using a Perkin Elmer Lambda 750 UV-VIS-NIR spectrometer in the range of 300–1,000 nm. The X-ray photoelectron spectroscopy spectra of P(Py:DSA:Py) films that were *in situ* polymerized on glass were also recorded to estimate the charge on the nitrogen atom of the Py using an Electron Spectroscopy for Chemical Analysis instrument (VG Multilab 2000). For X-ray diffraction measurements, thick PPy films (>1 $\mu$m) were prepared via multiple rounds of polymerization on glass substrates. The X-ray diffraction studies were performed using a Rigaku D/max-2500 diffractometer with Cu-K $\alpha$ radiation ($\lambda$ = 1.54 Å) at 40 kV and 100 mA. A high-voltage electron microscope (HVEM, JEOL Ltd., JEM ARM 1300S) with a point resolution of 0.12 nm was used to observe the polymer morphologies, and HRTEM images were recorded at 1,250 kV (Gatan Inc., SP-US1000HV). Sample specimens were prepared through the *in situ* polymerization of TMCPs on carbon-coated copper grids (200 mesh, EM Science) under the same reaction conditions used in the polymerization process for the formation of P(Py:DSA:Py) films on glass. The samples were coated very thinly on

the grids to allow electron beam transmission. The sheet resistances of PPy films ($\sim 150$ nm) formed via *in situ* polymerization on glass substrates ($1 \times 1$ cm$^2$ square in shape) were measured using a van der Pauw measurement[36,37] system from Bio-Rad Inc. Four electrode tips were placed in contact with the edges of the film to apply a current and detect a voltage. The mean value of the electrical conductivity for each polymer film was calculated from the sheet resistances of three samples measured at room temperature. The thicknesses of the films used for the conductivity measurements were measured using a SURFCORDER thickness profilometer.

**Data availability.** The data that support the findings of our study are available from the corresponding author on request.

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

## Acknowledgements

This work was supported by the Samsung Research Funding Center of Samsung Electronics under Project Number SRFC-MA1502-11.

## Author contributions

H.-J.L. designed and performed most of the experiments; H.-J.L and S.K. performed the spectroscopic characterizations; S.J.Y., J.-G.K. and Y.-J.K. performed the HVEM measurements and characterizations; H.-J.L., Y.-R.J. and B.-J.K. analysed the HRTEM and FFT data. All authors contributed to the discussion of the results and the preparation of the manuscript. J.-S.L. coordinated and directed the research project.

## Additional information

**Competing financial interests:** The authors declare no competing financial interests.

