## [Peer review file · Nature Communications]

Reviewers' comments:

Reviewer #1 (Remarks to the Author):

The manuscript "3-dimensional close-packed polymer crystals from two-monomer-connected precursors" shows a new method for the preparation of highly crystalline conjugated polymer films using precursors in which 2 pyrrole units are linked through a connector. The connectors are the acids MSA, EDSA, BDSA and BPDSA. The polymerization is performed in the presence of ammonium persulfate.

Overall, I find the ideas of the manuscript highly interesting and novel and I think it would be suitable for Nature Communications. The manuscript is well-written and clear.

I would however ask the authors to address the following issues before the manuscript can be considered for publication:

Questions:

- 1) How exactly is the polymerization performed on the substrate? Is polymerization only happening at the substrates or also within and at the walls of the vials? The authors should cite relevant literature on this interfacial process.
- 2) The authors should provide data on the pure pyrrole polymerization using the oxidant - without the acid.
- 3) The authors write „..... exhibited redshifted absorption spectra.... because of the regular growth of the polymers from the TMCPs..." on page 4. This is an interesting statement since the polymerization will lead - under the conditions used - in charged polypyrrole backbones. To solve this issue the authors should prepare their films on conducting substrates. Cyclic voltammetry and In-situ spectroelectrochemistry will give information about the neutral and the charged species. Only based on these data the observed red-shifts can be properly understood and explained. The spectra given in Figure S4 are not very clear and meaningful. This will also help the authors to be able to make statements about the polymerization mechanism and the nature of the crystalline films. It would be interesting to have statements about the degree of polymerization.
- 4) The reviewer is not convinced that „the DSA links caused the Ppy to be conductive" (page 4 last line) Ppy becomes conducting due to the oxidative polymerization used.
- 5) The exact conditions of the van der Pauw method should be given. Errors of the measurements are needed as well.

[redacted]

- 7) Figure S2: It is really hard to distinguish the bands in the FT-IR spectra.

Reviewer #2 (Remarks to the Author):

This Group reported successful preparation of 3-dimensional polymer crystals by connecting two monomers with different disulfuric acid linkers and polymerizing the precursors. The formation of 3-D polymer crystals were confirmed by various analytical techniques of XRD, HRTEM, and FFT images. The film of resulting polymers demonstrated flexibility on the various substrates. The work is well executed and the manuscript is also written very well, which makes this work above average standards.

However, the present work is very similar to work reported by author himself in the "Advanced Materials, 2012, 24, 3253-3257". Author has earlier reported that polymers containing pyridine can be cross-linked by dibromoalkane which forms 3-D molecular level ordering. In the current manuscript, author has reported formation of crystal using same strategy, and the previous work has not been cited. Author has not discussed any performance improvement over the previous work. It suggests that this manuscript lags significantly in originality. In addition, manuscript has some technical flaws, which needed to be addressed before the work is submitted to any journal.

(1) The synthesis of P(Py:MSA) and P(Py:DSA:Py) films by in-situ chemical oxidative polymerization has not been described completely as well as purification and characterizations especially. In detail, after polymerization, they characterized the polymers by only FT-IR, UV-Vis, and XPS. However, for investigation of polymer characteristics, other characterization such as NMR spectra, DSC, and TGA, etc are very important and needed to be reported.

(2) In Figure 4, depending on linker structure, conducting was influenced highly. However, if the polyaniline homopolymers and their conductivity were measured together as a control sample, the conductivity of the resulting polymer films in this study could be understandable more.

As a result, this study has no priority so that the contents are not suitable for publication in Nature Communications.

Reviewer #3 (Remarks to the Author):

Nice report about polymer-chain design. Overall, the manuscript shows novelty and excited result toward molecular-level crystallinities. In addition, the manuscript addresses in-depth analysis to support the content. It is well known that the crystallization behavior of the conducting polymers is the foundation in studying their intrinsic properties; further, obviously, these results can heighten properties of relevant devices. I strongly support this manuscript. I have only a comment about the using of "three dimensional". There is a little confusion of the concept of 3D but not too. I suggest that the authors change it if acceptable.

Responses to the comments and suggestions from the reviewers

(Manuscript ID: NCOMMS-16-07034-T)

Reviewer #1 (Remarks to the Author):

We greatly appreciate the valuable comments (shown in *italic* font) and have provided responses (the highlighted sentences can be found in the revised manuscript).

The manuscript "3-dimensional close-packed polymer crystals from two-monomer-connected precursors" shows a new method for the preparation of highly crystalline conjugated polymer films using precursors in which 2 pyrrole units are linked through a connector. The connectors are the acids MSA, EDSA, BDSA and BPDSA. The polymerization is performed in the presence of ammonium persulfate.

Overall, I find the ideas of the manuscript highly interesting and novel and I think it would be suitable for Nature Communications. The manuscript is well-written and clear. I would however ask the authors to address the following issues before the manuscript can be considered for publication:

Comment 1.

How exactly is the polymerization performed on the substrate? Is polymerization only happening at the substrates or also within and at the walls of the vials? The authors should cite relevant literature on this interfacial process.

Response:

We thank the reviewer for the valuable time spent in reviewing the manuscript. The polymerization simultaneously occurred 1) on the substrates, 2) in the solution (within the

vials) and 3) on the walls of the vials (film state). However, it was not feasible to characterize the polymer precipitates that formed during polymerization (in the solution) or the product that formed on the walls of the vials. This type of polymerization on various substrates has been previously reported by others [1-3].

A new sentence has been added on page 4, and the relevant literature has been cited; these new citations can be found in the revised manuscript as references #21-23 in “References” section.

Revised manuscript: (page 4, line 7)

“This type of *in situ* polymerization on various substrates has been previously reported by others²¹⁻²³,”

Revised manuscript: (page 10, line 20 in “References” section)

21. Zhang, H. & Li, C. Chemical synthesis of transparent and conducting polyaniline-poly(ethylene terephthalate) composite films. *Syn. Met.* **44**, 143–146 (1991).
22. Ferenets, M. & Harlin, A. Chemical in situ polymerization of polypyrrole on poly(methyl methacrylate) substrate. *Thin Solid Films* **515**, 5324–5328 (2007).
23. Chiou, N.-R., Lu, C., Guan, J., Lee, L.J. & Epstein, A.J. Growth and alignment of polyaniline nanofibres with superhydrophobic, superhydrophilic and other properties. *Nat. Nanotech.* **2**, 354–357 (2007).

Comment 2.

The authors should provide data on the pure pyrrole polymerization using the oxidant - without the acid.

Response:

Scheme 1 | Chemical oxidative polymerization of pure pyrrole.

We performed the polymerization of pure pyrrole using only the oxidant (**Scheme 1**) while maintaining all other conditions exactly the same as those used when synthesizing the polymer from two-monomer-connected precursors (TMCPs).

Figure 1 | UV-Vis absorption spectrum (a) and XRD spectrum (b) of PPy-oxidant. The thickness of the film used for UV-Vis spectroscopy was 150 nm. For the XRD measurements, a thick polymer film was prepared via multiple rounds of polymerization on glass substrates.

We observed that the resultant polymer possessed similar optical and electrical properties to those that have been reported by others in the case of pure polypyrrole (PPy-oxidant) synthesized via oxidative polymerization [4,5]. **Figure 1a** shows the UV-Vis spectrum of PPy-oxidant, which indicates that the oxidant causes the polymer to become partially doped [5] with sulfate counter anions, and the XRD analysis results (**Figure 1b**) show the amorphous nature of the material.

Table 1. The sheet resistances and conductivities of PPy-oxidant, P(Py:MSA) and P(Py:EDSA:Py).

Sample	Thickness	Sheet resistance (Ω/sq)	Conductivity (S/cm)
PPy-oxidant	~150 nm	8.26×10^4	0.806
P(Py:MSA)	~150 nm	2.18×10^4	3.04
P(Py:EDSA:Py)	~150 nm	7.44×10^3	8.94

We also compared the electrical conductivity of PPy-oxidant with those of P(Py:MSA) and P(Py:EDSA:Py) and observed that it was dramatically reduced because of the absence of the protonic acid dopant (**Table 1**).

Comment 3.

The authors write „..... exhibited redshifted absorption spectra.... because of the regular growth of the polymers from the TMCPs..." on page 4. This is an interesting statement since the polymerization will lead - under the conditions used - in charged polypyrrole backbones.

To solve this issue the authors should prepare their films on conducting substrates. Cyclic voltammetry and In-situ spectroelectrochemistry will give information about the neutral and the charged species. Only based on these data the observed red-shifts can be properly understood and explained. The spectra given in Figure S4 are not very clear and meaningful. This will also help the authors to be able to make statements about the polymerization mechanism and the nature of the crystalline films. It would be interesting to have statements about the degree of polymerization.

Response:

We thank the reviewer for this valuable suggestion. Regular growth in crystalline polymers such as ours can result in redshifts in the absorption spectra, as previously reported

by others [6-10]. The electronic interaction between the stacked polymer chains in a crystalline conductive polymer causes its energy band gap to be narrower compared with that of an amorphous polymer, although the polymer backbone structure is the same in both cases. Moreover, if the polymer forms a crystalline structure through the interdigitation of alkyl side chains or a more planar conformation of the backbone, redshifted absorption can occur even if the polymer has a charged structure [11]. Therefore, the P(Py:DSA:Py)s exhibited redshifted absorption spectra compared with that of the reference polymer, i.e., P(Py:MSA).

Figure 2 | Cyclic voltammograms of dedoped PPy (a), P(Py:MSA) (b) and P(Py:EDSA:Py) (c). A dedoped PPy film was prepared by immersing a PPy-oxidant film synthesized on ITO glass in a 1 M NaOH aqueous solution and rinsing it with water and ethanol. The cyclic voltammetry measurements were performed in an electrolyte of 1 M NBu₄PF₆ in acetonitrile with an Ag/AgCl reference electrode and a Pt wire as the counter electrode. NBu₄PF₆: tetrabutylammonium hexafluorophosphate.

We performed cyclic voltammetry on the ITO surface, as shown in **Figure 2**, to confirm the neutral or charged nature of P(Py:MSA) and P(Py:EDSA:Py). Oxidation and reduction peaks were observed for dedoped PPy because it is neutral. However, in the cases of P(Py:MSA) and P(Py:EDSA:Py), these peaks were absent and the current densities were higher because of their polaron and bipolaron states, thereby indicating that P(Py:MSA) and P(Py:EDSA:Py) have doped (oxidized or charged) structures. This behaviour has been previously reported [12].

Figure 3 | UV-Vis absorption spectra. a, UV-Vis spectra of dedoped PPy, P(Py:MSA) and P(Py:EDSA:Py). The measured polymer films were coated on optically clear glass substrates via *in situ* polymerization for 12 hours. UV-Vis spectra of P(Py:MSA) (b) and P(Py:EDSA:Py) (c) films after different reaction times early in the polymerization.

We also analysed the UV-Vis spectra of the polymers (dedoped PPy, P(Py:MSA) and P(Py:EDSA:Py)) synthesized on glass substrates (**Figure 3a**), and the results further confirmed the neutral and charged natures of the films. The neutral PPy exhibited an absorption maximum at ~350 nm, which is related to the π - π^* transition. By contrast, P(Py:MSA) and P(Py:EDSA:Py) showed redshifted absorption spectra because of the polaron or bipolaron bands due to the doping of the PPy with the sulfonic acids. Although MSA is more acidic than EDSA, the absorption band of P(Py:EDSA:Py) was redshifted compared with that of P(Py:MSA) because of the high crystallinity of P(Py:EDSA:Py) [6-11].

Instead of performing *in situ* spectroelectrochemistry, we measured the UV-Vis spectra of P(Py:MSA) and P(Py:EDSA:Py) films after various time intervals during polymerization (10 min, 30 min, 1 hour, and 2 hour), as shown in **Figures 3b and 3c**. The polymer films exhibited the characteristic absorption spectra of oxidized or doped PPy after 10 minutes, when the absorption spectra had already become redshifted compared with that of neutral polypyrrole, signifying that the polymer had already acquired the charged state. This type of behaviour has previously been reported with regard to *in situ* doping polymerization in the presence of organic acid [13]. We attempted to dissolve our polymer in various other solvents to determine the resulting degree of polymerization via GPC or ^1H NMR, but the polymer is nearly insoluble in any organic solvent. We are currently working on this issue to determine the molecular weight of the polymer.

The explanation of the relationship between the UV-Vis absorption spectrum and the nature of the crystallinity of the polymer has been modified in the revised manuscript, and new references have been added (references #26-28 in the “**References**” section in the revised manuscript).

Original manuscript: (page 4, line 18)

“P(Py:EDSA:Py), P(Py:BDSA:Py) and P(Py:BPDSA:Py) exhibited redshifted absorption spectra compared with that of P(Py:MSA) because of the regular growth of the polymers from the TMCPs.”

Revised manuscript: (page 4, line 20)

“Interestingly, P(Py:EDSA:Py), P(Py:BDSA:Py) and P(Py:BPDSA:Py) exhibited redshifted absorption spectra compared with that of P(Py:MSA). Such redshifted absorption indicates the existence of an electronic interaction between the stacked polymer chains, which causes the energy band gap to be narrower than that of an amorphous polymer with the same polymer backbone structure²⁶⁻²⁸.”

Revised manuscript: (page 11, line 6 in “References” section)

26. Nakano, T. Synthesis, structure and function of π -stacked polymers. *Polym. J.* **42**, 103–123 (2010).
27. Chen, M.S. *et al.* Enhanced solid-state order and field-effect hole mobility through control of nanoscale polymer aggregation. *J. Am. Chem. Soc.* **135**, 19229–19236 (2013).
28. D'Aprano, G. & Leclerc, M. Synthesis and characterization of polyaniline derivatives: poly(2-alkoxyanilines) and poly(2,5-dialkoxyanilines). *Chem. Mater.* **7**, 33–42 (1995).

Comment 4.

The reviewer is not convinced that „the DSA links caused the Ppy to be conductive” (page 4 last line) Ppy becomes conducting due to the oxidative polymerization used.

Response:

We thank the reviewer for pointing out this error. This statement was made in the

context that the conductivities of the doped polymers (doped with protonic acids such as sulfonic acids) were higher than that of the polymer synthesized via oxidative polymerization using only the oxidant [4,14]. Therefore, the relevant sentence has been modified in the revised manuscript.

Original manuscript: (page 4, line 24)

“This finding confirmed that all Py units were connected and that the DSA links caused the PPy to be conductive.”

Revised manuscript: (page 5, line 3)

“This finding confirmed that the Py units were linked by DSA connectors.”

Comment 5.

The exact conditions of the van der Pauw method should be given. Errors of the measurements are needed as well.

Response:

The van der Pauw (VDP) method is a conventional method for evaluating the electrical properties of semiconductor materials, such as resistivity, carrier density, and mobility [15,16]. However, we measured only the sheet resistances of our samples using this technique. The VDP method can be used to measure samples of arbitrary shape, although several basic sample conditions must be satisfied to obtain accurate measurements and avoid errors; for example, the thickness of the sample must be constant, the measurements must be performed by means of point contacts placed at the edges of the sample (as shown in **Figure 4**), and the sample quality must be homogeneous. Because the polymer chains in a conducting polymer are randomly distributed, the VDP method is most suitable in such cases and may provide

more reliable data than any other methods, such as the four point-probe method and the four line-probe method, in measurements of the sheet resistance.

Figure 4 | Schematic illustration of the VDP method. Four electrode tips were placed in contact with the edges of the polymer film, which was synthesized on a glass substrate (size: 1×1 cm²). R_s: Sheet resistance.

The sheet resistance (R_s) of the film is calculated from the voltage difference (V) between electrodes C and D when a constant current (I) is applied to electrodes A and B . To measure the exact resistances, it is important for the four contact positions to be on the edges of a square-shaped film because the error of this method depends on the contact positions of the electrodes [16]. Therefore, we measured the sheet resistances of 3 samples (film size: 1×1 cm²) for each polymer, placing the electrodes on the edges of the films, and calculated the average and deviation. These results were presented in Figure 4 of our original manuscript.

Detailed explanations of the VDP method and the results have been added to the revised manuscript and Supplementary Information, along with new references (references #36 and 37 in the “**References**” section in the revised manuscript).

Original Supplementary Information: (page 3, line 2)

“The sheet resistances of PPy films (~150 nm) formed via *in situ* polymerization on glass substrates were measured using a van der Pauw measurement system from Bio-Rad Inc. Four

electrode tips were placed in contact with the edges of the film to apply a bias.”

Revised Supplementary Information: (page 3, line 2)

“The sheet resistances of PPy films (~150 nm) formed via *in situ* polymerization on glass substrates (1×1 cm² square in shape) were measured using a van der Pauw measurement^{36,37} system from Bio-Rad Inc. Four electrode tips were placed in contact with the edges of the film to apply a current and detect a voltage. The mean value of the electrical conductivity for each polymer film was calculated from the sheet resistances of 3 samples measured at room temperature.”

Revised manuscript: (page 12, line 5 in “References” section)

36. van der Pauw, L.J. A method of measuring the resistivity and Hall coefficient on lamellae of arbitrary shape. *Philips Tech. Rev.* **20**, 220–224 (1958).

37. Banaszczyk, J., Schwarz, A., De Mey, G. & Van Langenhove, L. The Van der Pauw method for sheet resistance measurements of polypyrrole-coated para-aramide woven fabrics. *J. Appl. Polym. Sci.* **117**, 2553–2558 (2010).

The caption of Figure 4 has also been modified.

Original manuscript: (page 16, line 2)

“**Figure 4 | Electrical conductivities of P(Py:MSA) and P(Py:DSA:Py) films.** The electrical conductivities of the synthesized polymer films measured using the van der Pauw method at room temperature. (Inset: the average sheet resistance values of each film.)”

Revised manuscript: (page 17, line 2)

“**Figure 4 | Electrical conductivities of P(Py:MSA) and P(Py:DSA:Py) films.** The electrical conductivities of the synthesized polymer films measured using the van der Pauw

method at room temperature. The bar graphs show the mean conductivity measured for 3 samples of each polymer, with error bars representing the standard deviation. (*Inset: the average sheet resistance values of each film.*)”

Comment 6.

[redacted]

Response:

[redacted]

Comment 7.

Figure S2: It is really hard to distinguish the bands in the FT-IR spectra.

Response:

We thank the reviewer for this observation. In the revised manuscript, Supplementary Figure 2 has been modified as follows.

Original Supplementary Information: (page 5, line 1)

Revised Supplementary Information: (page 5, line 1)

References

- [1] Zhang, H. & Li, C. Chemical synthesis of transparent and conducting polyaniline-poly(ethylene terephthalate) composite films. *Syn. Met.* **44**, 143–146 (1991).
- [2] Ferenets, M. & Harlin, A. Chemical in situ polymerization of polypyrrole on poly(methyl methacrylate) substrate. *Thin Solid Films* **515**, 5324–5328 (2007).
- [3] Chiou, N.-R., Lu, C., Guan, J., Lee, L.J. & Epstein, A.J. Growth and alignment of polyaniline nanofibres with superhydrophobic, superhydrophilic and other properties. *Nat. Nanotech.* **2**, 354–357 (2007).

- [4] Chitte, H.K., Bhat, N.V., Gore, A.V. & Shind, G.N. Synthesis of polypyrrole using ammonium peroxy disulfate (APS) as oxidant together with some dopants for use in gas sensors. *Mater. Sci. Appl.* **2**, 1491–1498 (2011).
- [5] Ullah, H., Shah, A.A., Bilal, S. & Ayub, K. Doping and dedoping processes of polypyrrole: DFT study with hybrid functionals. *J. Phys. Chem. C* **118**, 17819–17830 (2014).
- [6] Houk, K.N., Lee, P.S. & Nendel, M. Polyacene and cyclacene geometries and electronic structures: Bond equalization, vanishing band gaps, and triplet ground states contrast with polyacetylene. *J. Org. Chem.* **66**, 5517–5521 (2001).
- [7] Nakano, T. Synthesis, structure and function of π -stacked polymers. *Polym. J.* **42**, 103–123 (2010).
- [8] Shi, M.-M. *et al.* π - π interaction among violanthrone molecules: Observation, enhancement, and resulting charge transport properties. *J. Phys. Chem. B* **115**, 618–623 (2011).
- [9] Chen, M.S. *et al.* Enhanced solid-state order and field-effect hole mobility through control of nanoscale polymer aggregation. *J. Am. Chem. Soc.* **135**, 19229–19236 (2013).
- [10] Wang, J. *et al.* Monodisperse macromolecules based on benzodithiophene and diketopyrrolopyrrole with strong NIR absorption and high mobility. *J. Mater. Chem. C* **4**, 3781–3791 (2016).
- [11] D'Aprano, G. & Leclerc, M. Synthesis and characterization of polyaniline derivatives: poly(2-alkoxyanilines) and poly(2,5-dialkoxyanilines). *Chem. Mater.* **7**, 33–42 (1995).
- [12] Flores-Estrella, R.A., Pacheco, D.E., Aguilar-Vega, M.J. & Smit, M.A. Study of the influence of a mixed electrolyte of oxalic acid and DBSA on the properties of co-deposited poly(aniline-co-pyrrole). *Int. J. Electrochem. Sci.* **3**, 1065–1080 (2008).
- [13] Shen, Y. & Wan, M. Tubular polypyrrole synthesized by in situ doping polymerization

- in the presence of organic function acids as dopants. *J. Polym. Sci. A Polym. Chem.* **37**, 1443–1449 (1999).
- [14] Shen, Y. & Wan, M. In situ doping polymerization of pyrrole with sulfonic acid as a dopant. *Syn. Met.* **96**, 127–132 (1998).
- [15] van der Pauw, L.J. A method of measuring the resistivity and Hall coefficient on lamellae of arbitrary shape. *Philips Tech. Rev.* **20**, 220–224 (1958).
- [16] Banaszczyk, J., Schwarz, A., De Mey, G. & Van Langenhove, L. The Van der Pauw method for sheet resistance measurements of polypyrrole-coated para-aramide woven fabrics. *J. Appl. Polym. Sci.* **117**, 2553–2558 (2010).
- [17] Egerton R.F., Li, P., & Malac M., Radiation damage in the TEM and SEM. *Micron* **35**, 399-409 (2004).
- [18] Loretto M.H. & Smallman R.E. An assessment of high voltage electron microscopy (HVEM). An invited review. *Mater Sci Eng.* **28** 1-32 (1977).
- [19] Makin M.J. & Sharp J.V. An introduction to high-voltage electron microscopy. *J Mater Sci.* **3** 360-371 (1968).

Reviewer #2 (Remarks to the Author)

We greatly appreciate the valuable comments (shown in *italic* font) and have provided responses (the highlighted sentences can be found in the revised manuscript).

This Group reported successful preparation of 3-dimensional polymer crystals by connecting two monomers with different disulfuric acid linkers and polymerizing the precursors. The formation of 3-D polymer crystals were confirmed by various analytical techniques of XRD, HRTEM, and FFT images. The film of resulting polymers demonstrated flexibility on the various substrates. The work is well executed and the manuscript is also written very well, which makes this work above average standards.

However, the present work is very similar to work reported by author himself in the "Advanced Materials, 2012, 24, 3253-3257". Author has earlier reported that polymers containing pyridine can be cross-linked by dibromoalkane which forms 3-D molecular level ordering. In the current manuscript, author has reported formation of crystal using same strategy, and the previous work has not been cited. Author has not discussed any performance improvement over the previous work. It suggests that this manuscript lags significantly in originality. In addition, manuscript has some technical flaws, which needed to be addressed before the work is submitted to any journal.

As a result, this study has no priority so that the contents are not suitable for publication in Nature Communications.

Response:

The reviewer's concerns are understandable. There is no doubt that the current work was inspired by our previous publication (*Advanced Materials, 2012, 24, 3253-3257*), in which we showed that an amorphous polymer, poly(2-vinylpyridine) (P2VP), when cross-linked

with 1,4-dibromobutane, influences the crystallinity of P2VP, leading to sub-nanometre-level molecular ordering in the P2VP polymer chain; we have, indeed, cited this work (reference #17 in the “References” section in the original manuscript).

Figure 1 | Strategy difference between previous research and current research.

However, in the present work, the strategy is not similar as previous work, which is related to the crosslinking of P2VP with dibromobutane. In this approach, we have used the connected monomers as a precursor for polymerization rather than a single monomer (**Figure 1**). When the single monomer is polymerized to form a linear chain, the chain direction is very likely to be randomized at each connection site due to an entropic effect, and therefore resulting structures are immensely entangled. When polymerization proceeds with a two-monomer-connected precursor (TMCP), which is not with a single monomer, four reactive sites of a TMCP (two from each monomer) introduces a constraint on the propagation direction, and prevents randomly oriented growth in monomeric or oligomeric state during

the polymerization unlike single monomer case (**Figure 1**). Therefore, we produced highly crystalline conjugated polymers using TMCPs, yielding crystalline structures while inhibiting the entropically favourable chain entanglements. Our results regarding this novel polymerization strategy and morphological analysis in 3 dimensions can serve as the basis for new approaches to polymer crystallography.

On the basis of these merits compared with our previous work, we strongly believe that the current work has significant novelty and deserves publication.

Comment 1.

The synthesis of P(Py:MSA) and P(Py:DSA:Py) films by in-situ chemical oxidative polymerization has not been described completely as well as purification and characterizations especially. In detail, after polymerization, they characterized the polymers by only FT-IR, UV-Vis, and XPS. However, for investigation of polymer characteristics, other characterization such as NMR spectra, DSC, and TGA, etc are very important and needed to be reported.

Response:

We thank the reviewer for this suggestion. The synthesis and purification of the P(Py:MSA) and P(Py:DSA:Py) films via *in situ* chemical oxidative polymerization has already been described in the methods section of the manuscript in full detail on page 18.

The polymerization simultaneously occurred 1) on the substrate, 2) in the solution (within the vials) and 3) on the walls of the vials. To enable ¹H NMR measurements, we attempted to dissolve the polymer precipitates that formed in the solution, but as is known, polypyrrole does not easily dissolve in any organic solvent, especially in our case, in which it had a connected (cross-linked) structure; therefore, we were not able to perform this

characterization. However, the valuable structural information of our polymers (which NMR can provide) has been discussed in the manuscript by FT-IR, UV-Vis and XPS analyses.

Figure 2 | TGA and DSC curves of P(Py:MSA), P(Py:EDSA:Py) and P(Py:BPDSA:Py).

We investigated the thermal properties of polymer samples (P(Py:MSA), P(Py:EDSA:Py) and P(Py:BPDSA:Py)) that had been freshly synthesized using the same methodology described in the manuscript. The powder samples were obtained through the purification of the precipitates that remained in the vials after polymerization. **Figure 2a** shows the thermal gravimetric analysis (TGA) results for all polymers, which began to degrade above 200 °C, consistent with the values previously reported in the literature for polypyrrole [1]. As seen from the thermograms, the degradation temperature (T_d) showed no change among the samples, but as the ordering of the molecular structures of the connectors increased from EDSA to BPDSA, they showed relatively higher thermal stability compared with MSA (the control sample). In the case of P(Py:BPDSA:Py), the thermal stability was the highest among the polymers above ~500 °C; this is because of the aromatic rings in the BPDSA connector. **Figure 2b** shows the differential scanning calorimetry (DSC) results for the polymers; here, little difference is evident among the thermograms. Broad endothermic bands were observed in all samples. Such bands have been regarded as a glass transition temperature (T_g) in some

studies [2], although their nature is still not clear. However, they may be accompanied by the expulsion of residual water (<150 °C).

Comment 2.

In Figure 4, depending on linker structure, conducting was influenced highly. However, if the polyaniline homopolymers and their conductivity were measured together as a control sample, the conductivity of the resulting polymer films in this study could be understandable more.

Response:

As the reviewer suggests, it is very important to compare the electrical conductivities of our P(Py:DSA:Py)s with that of the homopolymer (control sample). Because this work is based on polypyrrole, we believe that it is not appropriate to compare the electrical conductivities of the P(Py:DSA:Py)s with those of polyaniline homopolymers. However, we have already discussed the differences in conductivity between the non-connected homopolymer (P(Py:MSA)) and the connected (cross-linked) polymers (P(Py:DSA:Py)s) in the original manuscript on page 7.

Table 1. The sheet resistances and conductivities of PPy-oxidant, P(Py:MSA) and P(Py:DSA:Py)s.

Sample	Thickness	Sheet resistance (Ω/sq)	Conductivity (S/cm)
PPy-oxidant	~150 nm	8.26×10^4	0.806
P(Py:MSA)	~150 nm	2.18×10^4	3.04
P(Py:EDSA:Py)	~150 nm	7.44×10^3	8.94
P(Py:BDSA:Py)	~150 nm	9.87×10^3	6.75
P(Py:BPDSA:Py)	~150 nm	6.43×10^2	103.7

Table 1 shows the conductivity of P(Py:MSA), P(Py:DSA:Py)s and PPy-oxidant (from the polymerization of pure pyrrole using oxidant without any acid). The electrical conductivities of P(Py:DSA:Py)s were higher than that of P(Py:MSA) due to their crystallinity. The conductivity of PPy-oxidant was dramatically reduced compared to other polymers because of the absence of the protonic acid dopant.

References

- [1] Panigrahi, R. & Srivastava, S.K. Trapping of microwave radiation in hollow polypyrrole microsphere through enhanced internal reflection: A novel approach. *Sci. Rep.* **5**, 7638 (2015).
- [2] Karambelkar, V.V., Ekhe, J.D. & Paul, S.N. High yield polypyrrole: A novel approach to synthesis and characterization. *J. Mater. Sci.* **46**, 5324–5331 (2011).

Reviewer #3 (Remarks to the Author)

We greatly appreciate the valuable comments (shown in *italic* font) and have provided responses (the highlighted sentences can be found in the revised manuscript).

Nice report about polymer-chain design. Overall, the manuscript shows novelty and excited result toward molecular-level crystallinities. In addition, the manuscript addresses in-depth analysis to support the content. It is well known that the crystallization behavior of the conducting polymers is the foundation in studying their intrinsic properties; further, obviously, these results can heighten properties of relevant devices. I strongly support this manuscript. I have only a comment about the using of "three dimensional". There is a little confusion of the concept of 3D but not too. I suggest that the authors change it if acceptable.

Response:

Thank you very much for appreciating our work. The term “three dimensional” has been removed from appropriate places in the manuscript where it initially appeared in relation to our polymers.

Original title:

“3-dimensional close-packed polymer crystals from two-monomer-connected precursors”

Revised title:

“Close-packed polymer crystals from two-monomer-connected precursors”

Original manuscript: (at page 3, line 8)

“yielding 3-dimensional molecular crystals”

Revised manuscript: (at page 3, line 8)

“yielding crystalline structures at the molecular scale”

Original manuscript: (at page 3, line 11)

“Here, the role of the dopant is to endow the 3-dimensional crystal structure with conductivity.”

Revised manuscript: (at page 3, line 11)

“Here, the role of the dopant is to endow the polymer crystal structure with conductivity.”

Original manuscript: (at page 5, line 1)

“3-dimensional molecular crystal structures in the P(Py:DSA:Py)s.”

Revised manuscript: (at page 5, line 6)

“Molecular crystal structures in the P(Py:DSA:Py)s.”

Original manuscript: (at page 7, line 16)

“These results suggest not only that the 3-dimensional crystal structure strongly affects the electrical conduction pathways”

Revised manuscript: (at page 7, line 21)

“These results suggest not only that the crystal structure strongly affects the electrical conduction pathways”

Original manuscript: (at page 8, line 2)

“Polymerization from a TMCP results in a considerable increase in electrical conductivity because of the 3-dimensional close-packed crystal structures”

Revised manuscript: (at page 8, line 6)

“Polymerization from a TMCP results in a considerable increase in electrical conductivity because of the close-packed crystal structures”

REVIEWERS' COMMENTS:**Reviewer #1 (Remarks to the Author):**

In my opinion the authors have replied in a comprehensive way and changed the manuscript accordingly. I would recommend acceptance of the manuscript.

Reviewer #2 (Remarks to the Author):

Dr. Lee and other authors gave very detail response about each comment at this moment significantly. As a result, this revised version of manuscript are suitable for publication in Nature Communications.

Responses to Reviewers' Comments

Reviewer #1: In my opinion the authors have replied in a comprehensive way and changed the manuscript accordingly. I would recommend acceptance of the manuscript.

Response to the reviewer 1: We thank the reviewer for their time spent in reviewing our manuscript really appreciate for recognizing the quality and importance of our work.

Reviewer #2: Dr. Lee and other authors gave very detail response about each comment at this moment significantly. As a result, this revised version of manuscript is suitable for publication in Nature Communications.

Response to the reviewer 2: We thank the reviewer for their time spent in reviewing our manuscript and really appreciate for recognizing the quality and importance of our work.